# From Physical Distancing to Social Loneliness among Gay Men and Other Men Having Sex with Men in Belgium: Examining the Disruption of the Social Network and Social Support Structures

**DOI:** 10.3390/ijerph19116873

**Published:** 2022-06-04

**Authors:** Estrelle Thunnissen, Veerle Buffel, Thijs Reyniers, Christiana Nöstlinger, Edwin Wouters

**Affiliations:** 1Centre for Population, Family and Health, University of Antwerp, 2000 Antwerpen, Belgium; veerle.buffel@uantwerpen.be (V.B.); edwin.wouters@uantwerpen.be (E.W.); 2Institute of Tropical Medicine, 2000 Antwerpen, Belgium; treyniers@itg.be (T.R.); cnostlinger@itg.be (C.N.)

**Keywords:** loneliness, social network, social support, social provisions, men who have sex with men, physical distancing, lockdown, SARS-CoV-2

## Abstract

Since the start of the SARS-CoV-2 pandemic, levels of loneliness have increased among the general population and especially among sexual minorities, such as gay men and other men who have sex with men, who already experienced more problems with social isolation before the pandemic. We analyzed how the disruption of the social network and social support structures by containment measures impact loneliness among gay and other men having sex with men. Our sample consisted of gay and other men having sex with men who had in person communication with family as well as heterosexual friends and homosexual friends before the lockdown (N = 461). Multivariate regression analyses were performed with social provisions (social interaction and reliable alliance) and loneliness as dependent variables. A change from in-person communication with gay peers before the pandemic to remote-only or no communication with gay peers during the pandemic, mediated by change in social integration, was related to an increased feeling of loneliness during the pandemic compared with before the pandemic. There were some unexpected findings, which should be interpreted in the specific social context of the SARS-CoV-2 pandemic. On average, social integration and reliable alliance among MSM increased during the lockdown, even though in-person communication decreased and loneliness increased. Our results show it is critical to maintain a view of social support and social loneliness as lodged within larger social and cultural contexts that ultimately shape the mechanisms behind them.

## 1. Introduction

### 1.1. Problem Statement

Even before the SARS-CoV-2 pandemic, the World Health Organization declared social disconnectedness a major public health crisis among the general population [1], with sexual minority populations being even more prone to social isolation and loneliness compared with the general population. This increased vulnerability of sexual minority populations is caused by minority stress, being more likely to live alone, and a high incidence of comorbid health problems [2].

The SARS-CoV-2 pandemic and its associated social distancing measures significantly increased overall levels of loneliness among the general population [1,3,4,5,6]. Studies show that sexual minority populations—such as gay men and men who have sex with men (MSM), who are the focus of this study—experienced even greater psychological distress in relation to physical distancing [7,8] and that they had more problems coping [9]. However, little attention has been paid to the mechanisms underlying the increased impact on sexual minority populations. The SARS-CoV-2 pandemic has been a case of forced social isolation on a scale bigger than ever before, which presented a unique case to re-examine the relationships between social isolation, social support, and loneliness [10].

Loneliness research among sexual minorities has predominantly focused on sub-populations at increased risk of loneliness such as the elderly, youth, and those with comorbid health problems or who at risk of acquiring HIV [11,12,13,14]. To complicate matters further, the individuals in the studies vary in their level of social isolation, their specific path towards social isolation, and their reasons for self-isolating.

Our objective is to test whether existing theories regarding social support and loneliness apply to a cross-section of a sexual minority population undergoing similar social isolation instead of only those subgroups already considered at a relatively higher risk. We also aim to assess whether the relationships between social isolation, social support, and social loneliness remain the same when social isolation is sudden and enforced, or whether the mechanism behind social support and loneliness is different under these conditions.

Finally, the sexual minority theory proposes that peer communication is of added importance among sexual minorities in providing a sense of belonging and social support and to prevent loneliness [15,16,17,18,19]. With changes in modes of communication from in-person to remote or no communication, we test whether sexual minority individuals perceive a decrease in social support and an increase in loneliness when they shift to remote-only communication or no communication with gay peers and whether this shift has more impact than the same shifts in communication with family and heterosexual peers.

### 1.2. Physical Distancing among LGBTQI+ during the SARS-CoV-2 Pandemic

Physical distancing is an external and disruptive shock to our social networks and social support structures that has been linked to loneliness. A systematic review found that, population wide, loneliness has increased during the current SARS-CoV-2 pandemic [20] with those under stay-at-home, shelter-in-place, lockdown, or community restrictions to socially isolate being significantly more lonely [5], while continued in-person interactions during these restrictions were found to be protective against loneliness [21]. While remote communication replaced in-person communication during the pandemic to a large extent, early research suggested that technologically mediated communication was not an effective substitute when it comes to preventing loneliness during physical distancing measures [22,23].

A shift to remote communication with peers may have a more negative impact on sexual minorities than on the general population. Research has demonstrated that social networks of gay and other men who have sex with men (MSM) are constructed differently than the general population; there is greater importance in peer communication for this group [16,17,18], with positive effects of in-person contact [17,18]. The availability of safe inclusive gathering places and events to meet gay peers in person is thought to be of particular importance for providing a sense of connectedness for gay and other men having sex with men [17,18]. Gay men and other men having sex with men are likely to rely on their gay peers for social support [18], and recent findings have shown that the majority feels most isolated from gay peers compared with other social ties during the pandemic [24].

As physical distancing measures have led to the closure of LGBTQI+ meeting places and cancelation of public events such as Gay Pride, this social support from peers may have been, and may still be, in jeopardy. The sudden rupture in the social-affective networks of gay and other men having sex with men may has potentially increased loneliness; however, we currently lack understanding of how this disruption of the social network and social support structures impacted loneliness among gay and other men having sex with men and whether this disruption is likely to have lasting effects.

### 1.3. Conditions for LGBTQI+ during Lockdown in Flanders, Belgium

Globally, governments reacted in different ways to the pandemic to reach different goals; from Sweden, where the public were expected to follow a series of non-voluntary recommendations, to locking down entire cities to prevent further spread such as China.

The Federal Government of Belgium (where our study takes place) aimed to ‘flatten the curve’, i.e., to keep the number of hospitalizations within the capacity of hospitals. On 18 March 2020, Belgium entered its first lockdown, which would last until the 4 May 2020. Citizens were ordered to stay at home, only going outside for ‘essential travel’, e.g., grocery shopping, visiting a doctor or apothecary, traveling to a job categorized as ‘essential’, and for exercise. All public amenities and commercial venues not designated as essential were forced to close. People were to minimize communication with other people, excluding only members of the household from this regulation. Anyone not working in a designated ‘essential sector’ was required to work from home, education had to take place online, and social gatherings were forbidden, both inside and outside. It was allowed to meet one non-household member outside, keeping 1.5 m apart. Meeting non-household members inside was not permitted [25].

In their report about 2020 ‘COVID: A test for Human Rights’, Unia, (an independent public institution that fights discrimination and promotes equal opportunities in Belgium) determined that the law and public media discourse regarding the lockdown took a heteronormative interpretation of the term ‘nuclear family’: ‘By associating the concept of safety with home, a strong image of the nuclear family was put forward, which marginalized other familial and emotional ways of life in public discourse’ [26]. The heteronormative interpretation of ‘nuclear family’, which excluded the ‘found family’ of LGBTQI+ peers, likely created an environment that was even more conducive to social isolation among sexual minorities than among the general population.

### 1.4. Theoretical Framework

#### 1.4.1. Social Loneliness

Various types of loneliness can be distinguished, but only social loneliness is conceptualized as able to increase or decrease in response to a change in society [27], e.g., physical distancing measures. Of the other types of loneliness, existential loneliness is considered a universal human characteristic, inborn in all persons and not related to object loss or lack of intimate relationships [28,29]. Pathological loneliness is thought to be related to dysfunctional cognitions and affective states in psychiatric patients [28]. Emotional loneliness occurs when an individual lacks an intimate relationship with one special person [30]. Finally, social loneliness occurs when an individual lacks friends or feels no sense of belonging to a community. In this paper we focus exclusively on social loneliness, which is often a consequence of situational change or temporary separation [27]. Social loneliness is characterized by feelings of boredom, aimlessness, and marginality. In other words, socially lonely people do not feel they ‘belong’ to a group [31].

#### 1.4.2. Remote Communication

Both the fear of infection and the containment measures impacted how we connected to others, in a group or as individuals, in person or remotely, and how our social worlds looked as a whole. One’s social world can be represented by the ego-centered network, which consists of an individual and everyone whom he or she knows and with whom she or he interacts on a regular basis [32]. This interaction with your social world can be face-to-face or mediated by technology. Following the SARS-CoV-2 outbreak, countries worldwide implemented containment measures intended to minimize face-to-face communication and promoted the use of remote communication as a substitute [33]. The effects of these changes in modes of communication on social isolation and loneliness remain uncertain. A review of three randomized clinical trials among older adults found no protective effect against loneliness from videoconferencing during the pandemic [34]. A study of social networks among the general population found that in-person communication was associated with a smaller increase in loneliness during the pandemic, while no effect was found for remote communication [21]. Likewise, Litwin and Lavinski [33] found that while in-person communication protected against negative mental health outcomes during the pandemic, remote communication slightly exacerbated these outcomes. There is a dearth of studies on how the shift to remote communication impacted sexual minorities. The pandemic presents an unusual opportunity to study whether remote communication can act as a substitute for in-person communication when it comes to providing social support for this population.

#### 1.4.3. Model of Social Provisions

One way of conceptualizing social support is through the model of social provisions designed by Weiss [35]. This model takes the social needs of the individual as the starting point. The specific social needs of an individual can be seen as the ‘functions’ of their communication with members of their social network. Based on his analysis of various social functions of interpersonal relationships, Weiss [35] proposed a model of six types of ‘social provisions’. He theorized that only a deficit in the social integration function would directly result in social loneliness; this was confirmed in subsequent empirical research [36,37,38,39,40].

Social integration is defined as: ‘the sense of belonging to a group that shares similar interests, concerns, and recreational activities’. With lockdowns closing focus points for regular and structured social activities such as schools, workplaces, and recreational facilities [21], the pandemic has likely been a very disruptive shock to the provision of social integration by the network [41]. In other words, as communication decreases or changes to remote only, deficits in people’s social integration needs become more likely and thus their social loneliness is likely to increase.

In such times of stress, the reliable alliance provision may have a buffering effect on mental health [35,40]. Reliable alliance is ‘the assurance that others can be counted upon for tangible assistance’ and is relevant to help people cope with negative feelings and consequences of external changes, for which the pandemic and lockdowns certainly qualify. However, while the need for reliable alliance may grow during the pandemic, providing tangible assistance to ties outside the household is difficult to combine with lockdown and sheltering regulations, potentially hindering people from providing and receiving reliable alliance; this would make them even more vulnerable to mental health risks.

#### 1.4.4. Type of Tie

Weiss [35] expected the different social provisions to be provided by contact with different ties (friends, family, etc.). They expected social integration to be fulfilled mainly by friends and reliable alliance mainly by family. There is reason to suspect this may be different for gay and other men having sex with men, who have been reported to assign greater importance to the ‘found family’ and peer communication [16,17,18]. Omoto and Snyder [42] further suggest that feeling connected to the LGBTQI+ community is an important provider for support provisions among sexual minorities.

While research has recently begun to examine the role of different sources of social support for gay and other men having sex with men, studies remain scarce. McConnel et al. [16] found that peer and family support frequently co-occur; those who experienced family support were also likely to receive peer support, while those who received peer support were not more likely to receive family support. In addition to considering the separate effects of different sources of social support, they suggest it is important to acknowledge the combination of sources of social support. With the large-scale disruptions of gay peer networks due to venue closure described by Camargo et al. [43] and the heteronormative focus of the Belgian lockdown regulation mentioned earlier, it may be that family fills the resulting gap in social needs, at least for those MSM not rejected by family.

### 1.5. Hypotheses

Based on the preliminary evidence linking in-person communication to prevention of loneliness and remote communication as a less effective substitute, we hypothesize that a reduction in communication (from in-person to no communication) or a change in mode of communication (from in-person to remote-only communication) with the ego-centered network will be related to an increase in social loneliness (**H1**).

Applying the model of social provisions to lockdown/sheltering in place, we hypothesize that a reduction of communication (from in-person to no communication) or change in mode of communication (from in-person to remote-only communication) with friends and family will be related to a reduction in social integration (a) and reliable alliance (b), respectively (**H2a and b**). A decrease in social integration (a) and reliable alliance (b) will be related to an increase in social loneliness (**H3a and b**). We expect that a change in the mode of communication with friends and family will have an indirect effect on social loneliness through the social integration and reliable alliance functions (**H4a and b**).

Based on findings about the importance of peer communication to create a sense of belonging among MSM, we hypothesize that the impact of in-person communication with gay peers is most strongly related to fulfilling social integration and reliable alliance provisions (a and b) and in preventing social loneliness (c) compared to in-person communication with other ties (**H5a, b and c**).

Acknowledging the importance of studying multiple sources of support, we test the hypothesis that the impact of losing in-person communication with gay friends on social integration (a), reliable alliance (b), and social loneliness (c) due to closure of LGBTQI+ venues is moderated by in-person communication with family members due to the heteronormative focus of the Belgian lockdown regulations. This hypothesis only applies to those MSM not rejected by family (**H6a, b and c**). For an overview of the hypotheses and visualization of how they are linked, see Figure 1.

## 2. Materials and Methods

### 2.1. Participants

In order to assess the above-cited hypotheses (also visualized in Figure 1) we use data on the social network, social provisions, and loneliness among MSM during the first Belgian lockdown. We used data from the ‘Corona and I’ study executed two weeks into the first Belgian lockdown. Inclusion criteria of the online survey were being 18 years or older, not exclusively heterosexual, and born in or living in Belgium. Eligible participants provided consent after having been informed about the study and its procedures.

The total survey sample consisted of 965 individuals who self-identify as LGBTQI+. As the sample consisted mainly of MSM (692 individuals), we chose to focus on MSM as the specific sexual minority studied. We intended to assess which ties (family, heterosexual friends, homosexual friends) supply social integration and reliable alliance. We also wanted to compare whether a shift to remote-only or no communication with homosexual friends has a bigger impact on loneliness compared to a similar shift regarding family and heterosexual friends. To accomplish this, we selected those MSM who had in-person communication with family as well as heterosexual friends and gay friends before the lockdown. This resulted in a sample of 461 gay and other men having sex with men (67% of MSM in the sample). The other 251 individuals either had in-person communication with no ties, one type of tie, or a combination of two types of ties. A detailed description of the total sample of LGBTQI+ was published in Reyniers et al., (2022) [44].

### 2.2. Procedures

The study was set up in collaboration with a Flemish sexual health organization (‘Sensoa’) and a lesbian, gay, bisexual, trans, queer, or intersex (LGBTQI) umbrella organization (‘çavaria’) and received ethical approval from the Institutional Review Board of the Institute of Tropical Medicine (Antwerp) and the Ethical Review Board of the University of Antwerp. Detailed data collection procedures were published elsewhere [44]. The study enabled us to answer the many calls for research on the mechanisms behind mental health consequences of the SARS-CoV-2 pandemic for gay and other men having sex with men [2,34,45,46,47,48,49,50].

### 2.3. Measures

#### 2.3.1. Mode of Communication

Communication before and during lockdown was measured by asking respondents ‘Who did you have communication with before/after March 18th?’ Answers were presented in a matrix containing type of ties (family, heterosexual friends, gay friends, roommates, steady partners, known non-steady partners, new or anonymous partners) and four answer options for mode of communication (‘in-person close’, ‘in-person distance’, ‘remote’, ‘no contact’). We use the variables family, heterosexual friends, and gay friends in our analysis. We reduced the answer categories to three by joining ‘in person keeping a 1.5-m distance’ and ‘in person closer than 1.5 m’ into one category ‘in person’, because ‘in person closer than 1.5 m’ had a count less than 5. We computed the category ‘remote only’ as those who did not have ‘in-person’ communication but did have ‘remote’ communication.

#### 2.3.2. Social Integration

Change in social integration was measured using a modified version of the corresponding subscale of the Social Provisions Scale [31]. Social integration was measured using the standard four items, of which two positively phrased and two negatively phrased and reverse scored. The items traditionally have four answer categories from ‘strongly disagree to strongly agree’, which are given scores from 1–4. The scale is computed by adding the scores of all four items. To keep the questionnaire short, we chose to ask about change from before to during the lockdown on this scale. Our intention was to avoid participants experiencing ‘respondent fatigue’ due to the repetitive nature of having to answer every question twice, for before and after the lockdown.

We asked participants to indicate how their situation had changed from before to during the pandemic. Our answer categories were: ‘this is more the case’, ‘this is the same’, ‘this is less the case’, with responses scored as 1, 0, −1. The total scale scores were obtained by summing the item scores. A score below zero indicates a decrease in the social provision and a score above zero indicates an increase, while zero indicates there was no change on this social provision.

The Cronbach’s alpha for the social integration subscale is poor (α = 0.4). However, the alphas may not reflect the reliability of this scale adequately, as it is a relatively simple measure of consistency between items that is also dependent on the number of items.

Furthermore, the alpha can specifically underestimate the reliability of a multidimensional scale measuring emotion such as the Social Provisions Subscales [51,52,53]. Our answer categories for the Social Provisions Subscales make the subscales even more multidimensional than they already are, by asking respondents to compare their feelings from before to during the lockdown, and this has likely impacted the alpha. Assessments of the Social Provisions Scale found typical Cronbach’s alpha of between 0.6–0.7 for the subscales [54,55]. We include sample statistics on the separate items in Table 1 to be transparent about the content of the subscale.

#### 2.3.3. Reliable Alliance

Reliable alliance was measured by three items with the same answer categories and scale computation as social integration. Of the items, one was positively phrased and two were negatively phrased and reverse scored. The fourth item of this subscale was erroneously not included in the French and Dutch versions of the questionnaire during translation. This resulted in missing scores on this item for most of the participants (as the survey took place in Belgium, where French and Dutch are the most spoken languages). For this reason, we chose to omit the fourth item from the scale. The scale does include the negatively phrased counter to the missing item. We included the reliable alliance scale in our analysis on the basis that no fundamental concept was thus missing from the scale. Cronbach’s alpha for the subscale indicated acceptable internal consistency (α = 0.6). We include sample statistics on the separate items in Table 2 to be transparent about the content of the subscale.

#### 2.3.4. Social Loneliness

Loneliness was measured using the UCLA 3-item loneliness scale [56], which has been the scale used most often to measure loneliness during the SARS-CoV-2 epidemic [57]. This scale comprises 3 questions that measure three dimensions of loneliness (relational connectedness). There were four response categories: ‘never to not at all’, ‘some days’, ‘more than half of the days’, and ‘almost all days’ with scores of 0 to 3. The scores were added together to give a range from 0 to 9.

A format was chosen to determine how the participants currently assessed their loneliness during the pandemic period compared with the period before the pandemic. We first asked participants to score the items for the period before the pandemic, retrospectively. Findings with retrospective bias are often considered inconsistent, as some people report more positively while others report more negatively on the past. However, recall biases, overestimation, or underestimation are on average canceled out, especially in subjective constructs such as loneliness [58].

After obtaining answers on loneliness before the pandemic, we then asked them to score the items again for how they were feeling now (which was during the first lockdown). The difference between scores was computed by subtracting pre-lockdown scores from those during the lockdown, resulting in a range from −9 (far less lonely during lockdown than before) to +9 (much lonelier during lockdown than before). Cronbach’s alpha indicated excellent internal consistency for the UCLA 3-item loneliness scale (α = 0.82 both before and during the lockdown). We include sample statistics on the separate items in Table 3 (as perceived before lockdown) and Table 4 (during lockdown) to be transparent about the content of the scale.

#### 2.3.5. Control Variables of Age and Economic Hardship

We limited the number of control variables to build parsimonious models, taking the relatively small sample size relative to the complexity of the models. We chose age and social economic hardship as these have been found to effect loneliness in an overwhelming amount of studies on mental health among the general population during the pandemic [4,59,60,61], as well as before the pandemic [62].

To obtain the variable economic hardship, we asked, ‘Which of these categories most adequately describes how you feel about your income?’ with a 5-point scale, with answers ranging from ‘I am very well off’ to ‘It’s very hard to make a living’. We created a dummy variable for economic hardship with everyone who answered either ‘It’s hard to make a living’ and ‘It’s very hard to make a living’ considered as experiencing economic hardship [63].

We dichotomized age (‘18 to 35′ and ‘36+’), because the literature shows that young adults had a higher risk of loneliness [20,46] during the pandemic, as well as because exploratory analyses showed significant higher loneliness for those 18 to 35, compared with those 36+.

### 2.4. Analysis

As the first step in the linear regression analyses, we took perceived change in reliable reliance and social integration as the dependent variables. We estimated two models: **Model 1a**, a baseline model to explore the effect of change in communication with family, heterosexual friends, and gay friends (testing **H2a,b and H5a,b**). In **Model 1b** we added interaction effects to test whether the effect of change in gay friend communication is moderated by change in family communication (testing **H6a,b**). 

As the second step in the analysis, **Model 2a**, we took change in loneliness as the dependent variable and included reliable alliance and social integration as predictors (testing **H1, H3a,b and H5c**), further adding the interaction effects in **Model 2b** (testing **H6c**).

In step three, **Model 3**, we estimated the effect of change in communication with gay friends, heterosexual friends, and family on social loneliness with social integration and reliable alliance as mediators of this relationship (testing **H4a,b**).

As our fourth and final step, **Model 4**, we estimated the effect of change in communication with gay friends, heterosexual friends, and family on social loneliness with reliable alliance as a moderator of the relationship between the social integration mediator and loneliness (testing **H4a,b**).

We used SPSS in combination with the PROCESS-macro [62] to perform the analyses.

## 3. Results

### 3.1. Social Integration

In **Model 1a**, we found that change to remote-only communication with gay friends (compared with in-person communication) was related to a decrease in social integration (b = −0.58 ***). This supports **H2a** that a change in mode of communication will be related to a reduction in social integration (a).

In **Model 1b** the interaction effect between ‘change in communication with family’ and ‘change in communication with gay peers’ was included. A change to remote-only communication with gay peers was strongly related to a decrease in social integration when people also changed to remote-only communication with their family (−0.73 **). A change to remote-only communication with gay friends was not significantly related to a decrease in social integration as long as they had continued in-person communication with their family. These findings support **H6a** that the impact of losing in-person communication with gay friends on social integration is moderated by in-person communication with family members during lockdown, for those MSM not rejected by family. See Table 5 for the full results of the linear regression.

### 3.2. Reliable Aliance 

In **Model 1a**, we found that change to remote-only communication with gay friends (compared with in-person communication) was related to a decrease in reliable alliance (b = −0.45 **). This supports **H2b** that a change in mode of communication will be related to a reduction in social integration (a) and reliable alliance (b). However, a change to remote-only communication with heterosexual friends (compared with in-person communication) was related to an increase in reliable alliance (b = 0.41 *), which contradicts **H2b**. During the lockdown, gay and other men having sex with men on average felt they could rely on their heterosexual friends less if they still saw them in person compared with remotely, whereas they felt they could rely on their gay friends more if they still saw them in person compared with remotely. Our findings support **H5a** that in-person communication with gay peers is most important for fulfilling the reliable alliance function compared with communication with heterosexual friends or family. 

In **Model 1b** the interaction between ‘change in communication with family’ and ‘change in communication with gay peers’ was included. The interaction terms were not significant and changes to remote-only communication with gay friends and heterosexual friends (compared with in-person communication) remained significant. These findings detract from **H6b** that the impact of losing in-person communication with gay friends on reliable alliance is moderated by in-person communication with family members. See Table 6 for the full results of the linear regression.

### 3.3. Social Loneliness

We performed a one sample t-test comparing scores of retrospectively perceived loneliness before the pandemic and loneliness scores during the pandemic. The difference between scores was significant; respondents reported more loneliness during the pandemic than before the pandemic.

In regression models **Model 2a** and **2b** we found that a decrease in social integration was related to an increase in loneliness perceived by respondents (**Model 2a** b = −0.34 ***; **Model 2b** b = −0.33 ***). **H3a** is supported by these results. See Table 7 for the full results of the linear regression.

In **Model 3** we model perceived difference in social loneliness scores on change in mode of communication, mediated by social integration and reliable alliance. In **Model 4** we model difference in social loneliness scores modeled on change in communication, mediated by social integration, and moderated by reliable alliance. **Models 3,4** contained significant estimates when using communication with gay friends as the predictor but not when using communication with heterosexual friends or family as the predictor. This supports **H5** that the impact of in-person communication with gay peers was most strongly related to fulfilling social integration and reliable alliance functions and preventing a perceived increase in social loneliness during lockdown compared with before lockdown.

In **Model 3** (presented in Figure 2) communication with gay friends had a significant effect on both mediators: social integration and reliable alliance. A change from in-person to remote-only communication with gay friends decreased social integration (b = −0.30 **) and indirectly affected loneliness (b = 0.11 *). See Table A1 in Appendix A for all indirect effects. Thus, **H4** was supported as change in communication with gay friends had an indirect effect on social loneliness through social integration and reliable alliance functions. A change from in-person communication to no communication decreased reliable alliance (b = −0.52 *) but did not directly or indirectly affect loneliness. Reliable alliance itself also did not affect loneliness; however, when mediated first by reliable alliance and then by social integration a change to no communication with gay friends did have an indirect effect on loneliness (b = 0.10 *).

In **Model 4** (presented in Figure 3) social integration was again estimated as a mediator, but reliable alliance was tested as a moderator of the relation between social integration and loneliness. Reliable alliance did not significantly moderate the relationship between social integration and loneliness. In this model, both changes to remote-only and no communication with gay friends (compared with in-person communication) were significantly related to social integration (b = −0.40 ** and b = −0.60 **, respectively). The negative relationship between increase in social integration and perceived increase in loneliness remained almost the same as in Model 3 (b = −0.36 ***). Both changes in mode of communication had a significant indirect effect on loneliness when mediated through social integration (change to remote only b = 0.14 and change to no communication b = 0.14, both at *p* < 0.05). Based on the results from Model 4 we conclude that that **H4a** is supported, while **H4b** is not. Change in mode of communication with gay friends had an indirect effect on social loneliness through the social integration function but not the reliable alliance function. See Table A2 in Appendix A for all indirect effects.

## 4. Discussion

### 4.1. The Relationship between Remote Communication and Loneliness

The implications of the change in modes of communication during the pandemic is a topic on which findings between studies diverge. In our study, we found that a shift from in-person communication with gay friends to remote-only communication increased perceived change in loneliness. This relationship was mediated by social integration. Kovacs [21], who analyzed networks among the general population, also reports that the only communication that protected significantly against loneliness was in-person communication with close ties. Grant et al. [64], who looked at the spatial component of loneliness, found that without meeting places the LGBTQI+ sense of belonging with local communities was significantly reduced during SARS-CoV-2. Socializing online paradoxically made these LGBTQI+ feel lonelier due to the lack of physical presence. Both these studies are in line with our results that in-person communication with gay friends is related to a smaller perceived increase in loneliness. On the other hand, Ellis and Dumas [65] found virtual time with friends related to higher depression but lower loneliness among adolescents. The relationship between mode of communication and loneliness may be different among different populations.

### 4.2. Increase in Social Integration and Reliable Alliance in the Initial Stages of the Pandemic

There were some unexpected findings that should be interpreted in the specific social context of SARS-CoV-2 pandemic. On average, social integration and reliable alliance among MSM increased during the lockdown, even though in-person communication decreased and loneliness was reported as higher during than before the pandemic.

A possible explanation for the unexpected increase in social integration may be that people feel more connected to society during a crisis. In their public campaign, the Belgian government emphasized the need to face the epidemic together, and grass-root initiatives to support vulnerable populations were widespread. The epidemic also provided a shared experience, which may have made it easier for people to relate to one another. A study into social support following the SARS epidemic in 2003 likewise showed improved social cohesion in Hong Kong after the pandemic and an intensive media coverage depicting ‘a more coherent and harmonious atmosphere’ in Hong Kong [66].

### 4.3. Exacerbation Mechanisms of Loneliness in the Context of the Pandemic 

The increase of loneliness perceived by respondents during the pandemic (despite an increase in social integration) may have been exacerbated by pandemic-specific conditions. De Jong Gierveld et al. [67] propose that ‘The intensity of loneliness is affected not only by the type of communications that are missed, but also by the time perspective required to “solve” and upgrade problematic relationships, and the capacities to change the situation’ (486). Loneliness may have intensified because of people’s inability to change the situation due to the physical distancing regulations.

Once loneliness is present in a network, it can also spread and reinforce itself. This may have happened on a grander scale during the pandemic due to the increase in loneliness as perceived by the population. Cacioppo et al. [68] found evidence of loneliness contagion within social networks; people who were in communication with social ties who were lonely were more likely to become lonely themselves and to spread this feeling to their other social ties. This mechanism was in place before the pandemic and, although in-person communication decreased, could continue to function through remote communication during the pandemic. The latter was described by Grant et al. [59], who found that respondents reported that online communication made them feel lonelier than before. As we also saw higher levels of loneliness during the pandemic compared with before, it seems likely that the amount of lonely people in MSM networks increased, which would set off loneliness contagion far more frequently than under normal circumstances. The contagion mechanism potentially continuously magnified the loneliness created by the pandemic, maybe reaching individuals who were not initially made lonely by the lockdown context itself.

### 4.4. The Danger of Loneliness that Has Passed: Network Shrinkage

Over time, decreased in-person contact can have serious and lasting effects on the social network. Roberts and Dunbar [69] found that feelings of closeness between friends and family members dropped by more than 30 percent after two months of no in-person contact. After five months, feelings of closeness with friends dropped close to 80 percent. This is in line with our findings that remote only and no contact decrease social integration. Less closeness, integration, and more loneliness during the pandemic may thus lead to social isolation not only during but also after. Cacioppo et al. [68] found that lonely people at the edges of the network cut ties with that network but not before transmitting these feelings of loneliness to their nearest ties in the network; leading to fraying from the outside inward. Thus, a lack of in-person contact with friends and accompanying feelings of loneliness during physical distancing may lead to substantial network shrinkage that outlasts the pandemic. Network shrinkage would be especially harmful for MSM who rely on in-person contact with their network of peers to provide feelings of social integration and identity affirmation.

### 4.5. The Pandemic and Other Types of Loneliness

It is possible that other types of loneliness, beside social loneliness, also increased during the pandemic. The UCLA 3-item loneliness scale is meant to measure social loneliness [52], but it is possible social loneliness, emotional loneliness, and existential loneliness were conflated in respondents’ answers, as these other types of loneliness were not measured. Emotional loneliness depends on the connection to a significant other [30], and as many non-cohabiting couples were physically separated during the first lockdown, this likely increased their emotional loneliness. Existential loneliness stems from the realization that a human being is fundamentally alone, with the accompanying emptiness, sadness, and longing [28,29]. It also seems likely that this realization is more prevalent and intense during a period of enforced physical and social isolation with high news coverage of loss of life and higher incidence of personal loss of loved ones. We recommend that future studies measure these types of loneliness separately but simultaneously, and study whether they can amplify each other.

### 4.6. Limitations

We collected data through an online survey, which makes a self-selection bias likely. The fact that we are working with a convenience sample, as well as a sample size which was relatively small compared with the intricacy of the models, limits the generalizability of the results. 

The survey was cross-sectional and asked after mode of communication and loneliness before the lockdown, retrospectively. Although it is likely that recall biases cancelled out on average loneliness before the pandemic remains vulnerable to under- or overestimation, it is especially vulnerable to overestimation due to the context of the pandemic. The implicit theory of change indicates that people recall former loneliness by combining their current state with their assumptions about how their loneliness has probably changed due to circumstances [58]. The context of the lockdown may have led people to assume their loneliness must have increased, thus reporting exaggerated low levels of loneliness before the lockdown.

In addition, the social integration and reliable alliance subscales are traditionally measured at one point in time, with respondents answering how much they agree with statements regarding their current situation [31]. Although we deliberately chose to modify the scales to ask after change from before to during the lockdown in order to prevent respondent fatigue, this is not how the scales were originally designed. A further limitation of the Reliable Alliance Subscale is that only three out of four items were measured, although the reversely phrased item of the missing item was included. A limitation of the Social Integration Subscale is that Cronbach’s alpha indicated poor internal consistency between items. The above may have impacted reliability of these traditionally robust scales.

The survey only contained questions about whether people were in communication with specific types of ties, not about the quantity and quality of those ties nor about the communication. We recommend future research include questions on these topics into their survey and recommend Kovacs et al. [21] as an excellent example.

We took a subsample of gay and other men having sex with men that had in-person communication with both family, gay friends, and heterosexual friends before the start of the pandemic because this was necessary to test the relative effects of change in the mode of communication. There is almost certainly a selection effect; therefore, those with more limited social networks, e.g., those more isolated (no in-person communication with one or more of the following: gay friends, heterosexual friends, or family) were excluded from the analysis.

### 4.7. Public Health Implications

Over time, loneliness past and present can have serious and lasting effects on the social network of MSM. A lack of in-person communication with peers and accompanying feelings of loneliness during several years of physical distancing may have led to substantial network shrinkage that likely outlasts the pandemic. Network shrinkage could be especially harmful for MSM who rely on in-person communication with their network of LGBTQI+ peers for social integration and identity affirmation.

Now that European countries have withdrawn most physical distancing regulations, and have ‘re-opened’, it is key to evaluate the degree to which pandemic-induced loneliness remains among MSM. The link between loneliness and network shrinkage also points to the importance of determining the degree of shrinkage of MSM peer networks and the current opportunities available to MSM to reform or re-expand their network. Lonely people are known to often withdraw further from social communication [67]. Even for those who never suffered from loneliness during the pandemic, or those who no longer do, re-integrating in peer networks may not be straightforward. We do not know how many LGBTQI+ venues have survived the epidemic and have re-opened, nor how many LGBQTI+ organizations have made the move back from online activities to activities that take place in person. The subject needs more study, but it seems likely to the authors that interventions centering around providing low-threshold opportunities for in-person peer communication would be beneficial to MSM mental health.

## 5. Conclusions

MSM who were in in-person communication with both family, heterosexual friends, and gay friends before the pandemic did not experience a decrease in social integration and reliable alliance during the first lockdown in Belgium; on the contrary, many indicated an increase. The public nature of the crisis may have made people feel more included in society. Alternatively, it may be that people experienced more social support because more situations arose in which it was both needed and provided. However, when MSM shifted to remote-only communication or no communication with gay friends, they perceived an in increase in loneliness, mediated by social integration. The same was not found for a shift to remote-only or no communication with family or heterosexual friends. This points to the importance of focusing MSM mental health interventions on providing opportunities for safe in-person communication with peers. Interventions should provide opportunities for MSM to connect with peers and grow their peer network to combat the loneliness that may be lingering from years of physical distancing.

## Figures and Tables

**Figure 1 ijerph-19-06873-f001:**
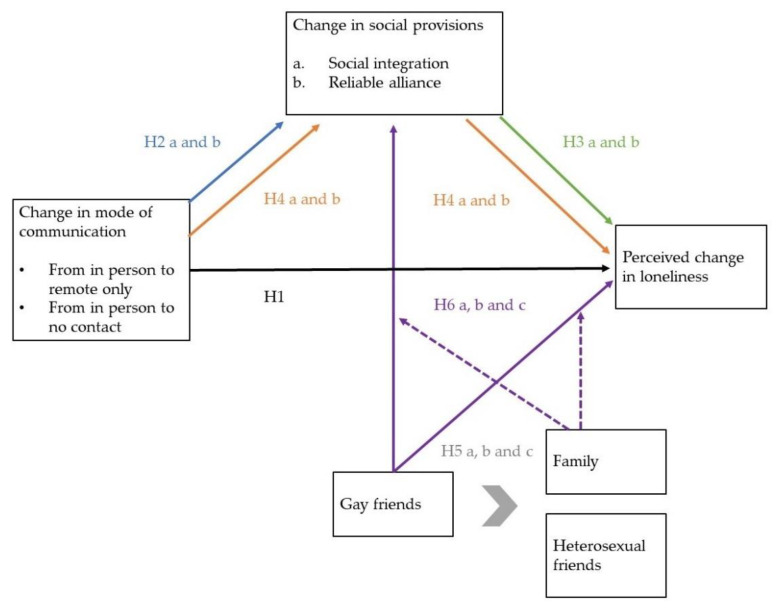
Conceptual model of the relationships between change in mode of communication, social functions, and loneliness. The unbroken lines represent connections between concepts, the dotted lines represent interaction effects between family and gay friends on the dependent variables.

**Figure 2 ijerph-19-06873-f002:**
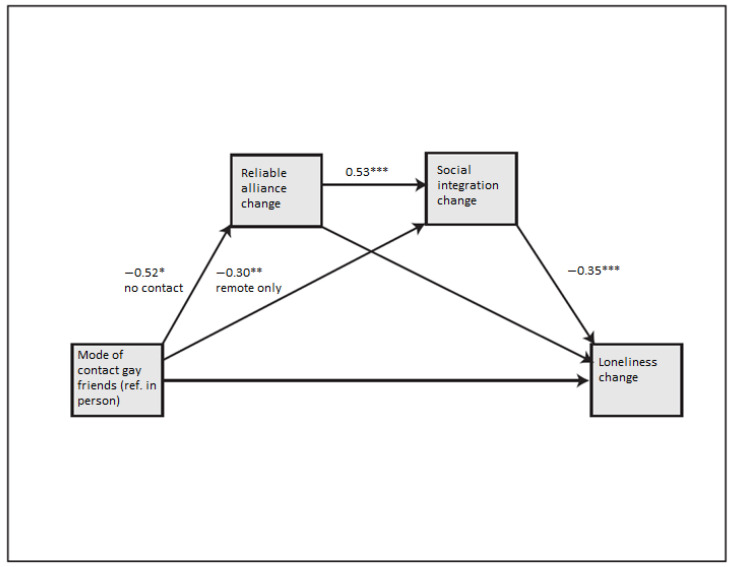
Model of the relationship between mode of communication with gay friends on difference in loneliness scores including reliable alliance change and social integration change as mediators. One * indicates a significance value of 0.05 or smaller, ** indicates a significance value of 0.01 or smaller, *** indicates a significance value of 0.001 or smaller.

**Figure 3 ijerph-19-06873-f003:**
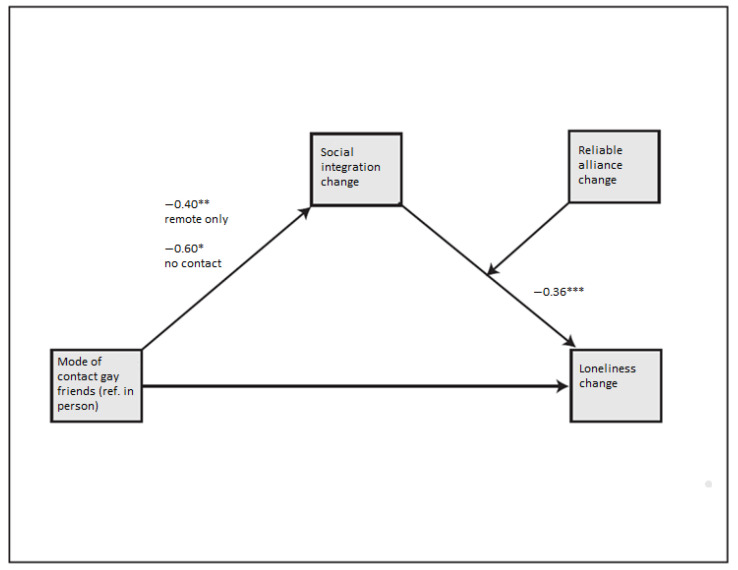
Model of the relationship between mode of communication with gay friends on difference in loneliness scores including social integration change as a mediator and reliable alliance change as a moderator. One * indicates a significance value of 0.05 or smaller, ** indicates a significance value of 0.01 or smaller, *** indicates a significance value of 0.001 or smaller.

**Table 1 ijerph-19-06873-t001:** Social integration subscale sample scores.

Item	Mean				Frequency		
	Statistic	Std. Error	Std. Deviation	Variance	‘This Is Less the Case’ (Score = −1)	‘This Is the Same’ (Score = 0)	‘This Is More the Case’ (Score = 1)
There are people who enjoy the same social activities as I do	2.16	0.03	0.58	0.33	46	296	119
I feel part of a group of people who share my attitudes and beliefs	2.05	0.02	0.47	0.22	40	358	63
There is no one who shares my interests and concerns	2.10	0.03	0.55	0.30	49	318	94
There is no one who likes to do the things I do	2.16	0.02	0.49	0.24	24	339	98

**Table 2 ijerph-19-06873-t002:** Reliable alliance subscale.

Item	Mean				Frequency		
	Statistic	Std. Error	Std. Deviation	Variance	‘This Is Less the Case’ (Score = −1)	‘This Is the Same’ (Score = 0)	‘This Is More the Case’ (Score = 1)
There are people I can count on in an emergency	2.10	0.02	0.47	0.22	31	354	76
There is no one I can depend on for aid if I really need it	2.23	2.23	0.53	0.28	25	307	129
If something went wrong no one would help me	2.26	2.26	0.56	0.32	29	283	149
There are people I can depend on to help me if I really need it	-	-	-	-	-	-	-

**Table 3 ijerph-19-06873-t003:** Perceived loneliness before the lockdown.

Item	Mean				Frequency			
	Statistic	Std. Error	Std. Deviation	Variance	‘Never to Not at All’ (Score = 0)	‘Some Days’ (Score = 1)	‘More than Half of the Days’ (score = 2)	‘Almost All Days’ (Score = 3)
How often do you feel that you lack companionship?	1.80	0.03	0.74	0.54	162	250	30	19
How often do you feel left out?	1.46	0.03	0.68	0.46	289	144	18	10
How often do you feel isolated from others?	1.58	0.04	0.77	0.60	255	164	23	19

**Table 4 ijerph-19-06873-t004:** Loneliness during the lockdown.

Item	Mean				Frequency			
	Statistic	Std. Error	Std. Deviation	Variance	‘Never to not at All’ (Score = 0)	‘Some Days’ (Score = 1)	‘More than Half of the Days’ (Score = 2)	‘Almost All Days’ (Score = 3)
How often do you feel that you lack companionship?	2.55	0.04	0.94	0.90	54	197	114	96
How often do you feel left out?	1.65	0.04	0.86	0.73	256	136	45	24
How often do you feel isolated from others?	2.41	0.05	1.00	1.00	88	182	104	87

**Table 5 ijerph-19-06873-t005:** Linear regression of change in social integration by age, economic hardship, and mode of communication per tie. The regression slope, also called unstandardized coefficient is represented by b, and the standard error of this coefficient by s.e. One * indicates a significance value of 0.05 or smaller, ** indicates a significance value of 0.01 or smaller, *** indicates a significance value of 0.001 or smaller.

	Model 1a		Model 1b	
	b	s.e.	b	s.e.
Age 18–35 (vs. 36+)	−0.07	0.13	−0.10	0.13
Economic hardship (ref. no economic hardship)	0.07	0.30	0.07	0.29
Gay friends, remote only (ref. in person)	−0.58 ***	0.16	−0.11	0.25
Gay friends, no communication (ref. in person)	−0.25	0.31	0.00	0.47
Heterosexual friends, remote only (ref. in person)	0.24	0.17	0.17	0.18
Heterosexual friends, no communication (ref. in person)	−0.49	0.27	−0.43	0.27
Family, remote only (ref. in person)	0.03	0.14	0.46 *	0.21
Family, no communication (ref. in person)	0.14	0.28	0.03	0.40
Gay friends, remote only × family, remote only (ref. in person)			−0.73 **	0.28
Gay friends, remote only × family, no communication (ref. in person)			0.50	0.76
Gay friends, no communication × family, remote only (ref. in person)			−0.83	0.73
Gay friends, no communication × family, no communication (ref. in person)			0.01	0.64

**Table 6 ijerph-19-06873-t006:** Linear regression of change in reliable alliance by age, economic hardship, and mode of communication per tie. The regression slope, also called unstandardized coefficient is represented by b, and the standard error of this coefficient by s.e. One * indicates a significance value of 0.05 or smaller, ** indicates a significance value of 0.01 or smaller.

	Model 1a		Model 1b	
	b	s.e.	b	s.e.
Age 18–35 (ref. 36+)	0.16	0.12	0.16	0.13
Economic hardship (ref. no economic hardship)	0.12	0.28	0.10	0.28
Gay friends, remote only (ref. in person)	−0.44 **	0.15	−0.49 *	0.24
Gay friends, no communication (ref. in person)	−0.22	0.30	−0.34	0.45
Heterosexual friends, remote only (ref. in person)	0.41 *	0.17	0.41 *	0.17
Heterosexual friends, no communication (ref. in person)	−0.14	0.26	−0.11	0.26
Family, remote only (ref. in person)	−0.05	0.13	−0.07	0.20
Family, no communication (ref. in person)	−0.20	0.26	−0.43	0.39
Gay friends, remote only × family, remote only (ref. in person)			0.06	0.27
Gay friends, remote only × family, no communication (ref. in person)			0.29	0.73
Gay friends, no communication × family, remote only (ref. in person)			−0.17	0.70
Gay friends, no communication × family, no communication (ref. in person)			0.39	0.61

**Table 7 ijerph-19-06873-t007:** Linear regression of loneliness by age, economic hardship-, mode of communication per tie, reliable alliance, and social integration. The regression slope, also called unstandardized coefficient is represented by b, and the standard error of this coefficient by s.e. *** indicates a significance value of 0.001 or smaller.

	Model 2a		Model 2b	
	b	s.e.	b	s.e.
Age 18–35 (ref. 36+)	0.28	0.20	0.30	0.20
Economic hardship (ref. no economic hardship)	0.47	0.46	0.45	0.46
Gay friends, remote only (ref. in person)	−0.31	0.25	−0.25	0.39
Gay friends, no communication (ref. in person)	−0.35	0.48	−0.89	0.73
Heterosexual friends, remote only (ref. in person)	0.24	0.27	0.24	0.27
Heterosexual friends, no communication (ref. in person)	−0.19	0.42	−0.29	0.42
Family, remote only (ref. in person)	−0.14	0.22	−0.21	0.33
Family, no communication (ref. in person)	−0.01	0.43	0.32	0.63
Reliable alliance	0.03	0.09	0.03	0.09
Social integration	−0.34 ***	0.09	−0.33 ***	0.09
Gay friends, remote only × family, remote only (ref. in person)			0.00	0.44
Gay friends, remote only × family, no communication (ref. in person)			−1.68	1.18
Gay friends, no communication × family, remote only (ref. in person)			1.42	1.14
Gay friends, no communication × family, no communication (ref. in person)			0.35	0.99

## Data Availability

Restrictions apply to the availability of these data. Data were obtained from the Institute of Tropical Medicine and are available from Estrelle Thunnissen with the permission of Thijs Reyniers.

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
