# Peer review of "From Physical Distancing to Social Loneliness among Gay Men and Other Men Having Sex with Men in Belgium: Examining the Disruption of the Social Network and Social Support Structures"

_ijerph, 2022, doi:10.3390/ijerph19116873_

Round 1
Reviewer 1 Report
From physical distancing to social loneliness among gay men and other men having sex with men in Belgium: examining the disruption of the social network and social support structures (ijerph-1660920)
The manuscript describes results from a cross-sectional study of a relatively large (N > 450) sample of Belgian gay men and other men who have sex with men who completed measures regarding social contacts and loneliness early in the COVID-19 pandemic. Results showed increases in loneliness especially among participants who reported no contact or remote contact only with gay friends, relative to individuals who reported in person contact. However, an unexpected finding was the social integration scores increased.
Strengths of the study include the investigation of an important, timely topic in a vulnerable population. Limitations include several aspects of the presentation of findings. In particular, there are several places in the manuscript where additional detail and additional clarity are needed.
Comments below may serve to strengthen the manuscript.
- The manuscript notes that 692 gay and other men who have sex with men were included in the sample but analyses were limited to the 461 individuals who reported in person contact with their social network prior to the pandemic. Just to clarify, 231 out of 692 participants (33%) reported no in person contact with their social networks prior to the pandemic? This percent seems quite high.
- Two of the categories of types of contact (“in person closer than 1.5 meters” and “in person keeping a 1.5 meter distance”) were collapsed into one because fewer than 5 (1% of the sample) reported being in the “in person closer than 1.5 meters” category. This seems like a very surprising number. Did only 1% of the sample live with a partner, or family, or a roommate, or were these social contacts excluded from the count in some way? Is it typical that only 1% of gay and men who have sex with men in Belgium live with other people? If not, how representative is this sample?
- Additional detail regarding the scales would aid the reader in understanding the findings. The Social Provisions Scale (SPS) was used to assess social integration and reliable alliance. The original measure has four items assessing each of these constructs. In this manuscript, 4 items are used to assess social integration and 3 items are used to assess reliable alliance. What was the rationale for shortening the reliable alliance scale? What was the internal consistency of the scales? The discussion notes that the internal consistency was low but the actual Cronbach’s alpha should be reported in the methods section. Providing a few sample items for each of these scales would aid the reader in understanding. Finally, the scale is incorrectly called “Social Provision Scale” in the manuscript.
- The modified scoring for the measures is difficult to follow. The manuscript states “The items remained the same, but answer option were modified to ‘this is more the case’, ‘this is the same’, ‘this is less the case’, with responses scored as 1, 0, -1. The total scale scores were obtained by summing the item scores coming to a range of -4 to +4 for social integration and -3 to +3 for reliable alliance.”
So, for these scales, a negative number means that overall, “this is less the case” during the pandemic, meaning lower social integration and lower reliable alliance?
- If a prior study or existing measure used the same method to assess socioeconomic position, it would be helpful to cite that source.
- The manuscript notes that age was controlled for in some analyses. It reports in the method section that two categories were used: “18 to 65” and “65+”. How many participants were in each group? What was the rationale for dichotomizing along this dimension?
Relatedly, Table 1 indicates two age categories “18-35” and “35+”. Table 2 compares two age categories “18-35” and “36+”. Table 2 would seem to be more logical as a 35 year old in Table 1 would appear in both age groups.
If “18-65” versus “65+”was used as a control, it would be preferable to report the percentages in these two categories in the tables. Alternatively, it may be that the reference to 18-65 and 65+ in the methods is incorrect? Clarification on this point is needed.
- Additional detail is needed for Table 2. A more descriptive title than “Bivariate results” would be helpful, as would an indication of the tests that were used and the actual value of those tests rather than simply an asterisk indicating significance. As it stands, I cannot interpret Table 2.
- The words “integration” and “interaction” are used multiple times throughout the manuscript. This is necessary as the study assessed social integration and conducted analyses looking at interactions between variables. There are some places where these words are transposed which is quite confusing. For example, “social interaction and reliable alliance” in the abstract, “Only for change in social integration an integration term proved significant” (line 272). Fixing these would increase the clarity of the manuscript.
- Both “corona-epidemic” and “SARS-COV2 pandemic” are used to refer to the pandemic. It would be preferable to pick a term and use that throughout the manuscript.
- The abstract should refer to “dependent” variables rather than “depending” variables.
Reviewer 2 Report
This is a really interesting paper concerning the effect of social isolation during the pandemic on a sexual minority population gay men and msm in Belgium.
Some recommendations for the form of the presentation of the paper:
- In the abstract, the results are not being adequately described and the reader does not get a clear picture of the findings while reading it.
- The introduction and the theoretical framework section of Materials and Methods could be merged ( since both refer to theoretical background) and perhaps shortened in terms of the parameters of social contact/communication/isolation that are being described.
- According to my view, the analytic presentation of many tables and model figures renders the understanding of concise results more loose and difficult and I would suggest some of the tables to be omitted and replaced by summarised tables with the main findings.
- In the discussion section, the following passage : "There were some unexpected findings, which should be interpreted in the specific social context of SARS-CoV-2 pandemic. On average social integration and reliable alliance among MSM increased during the lockdown, even though in person contact decreased and loneliness increased".ll 358-360", summarises in a concise way the contradictory but interesting findings of the paper and should be included in the abstract section.
- I suggest that the basic hypotheses and models should be summarised in the same sections of the text (Introduction/Discussion perhaps in two paragraphs in both sections) because as they become analysed in different paragraphs, they are more difficult to be followed in their interconnectivity.
Reviewer 3 Report
This paper has potential to make theoretical and practical contributions, but there are places that need further elaboration. Below, I would like to suggest a few points that might be helpful for future revision.
- Keywords: What does MSM stand for? Please use a full term for a keyword.
- Line 26: Please explain the reason why men who have sex with men are more prone to social isolation and loneliness. Are they even more prone to social isolation than women having sex with women?
- Line 54: Please explain how the understanding of the mechanism underlying social network disruption and loneliness fill the gap in the literature. A more detailed explanation of the study’s theoretical and practical contribution would be helpful.
- Line 57: The title ‘materials and methods’ does not seem to fit. My suggestion is ‘theoretical framework and hypotheses’.
- Line 81: ‘… before the’ This sentence is incomplete. Line 101 as well.
- Line 116: Please explain social provision theory. Is this an overarching theory of this study?
- Line 123: Isn’t it ‘social integration’ instead of ‘social interaction’?
- Line 143: What might be the logic behind H6 that family fills the gap in social needs? If the assumption is that gay men are often rejected by family, how would they procure family support during the pandemic?
- Line 185: Please provide sample measurement items for each variable (social integration, reliable alliance, loneliness). Also, measurement reliability coefficients need to be reported.
- Line 202: What are the reasons for controlling for the effect of age and social economic hardship?
- Please use period instead of comma for decimal points.
- Line 271: The meaning of the sentence ‘Only for change in social integration an integration term proved significant’ is not clear.
- Line 279: The sentence is incorrect. Decrease in social integration was related to an increase in social loneliness.
- Line 309: Please specify the figure number.
- Discussion: I would rather move the first paragraph to the end of the discussion section.
Reviewer 4 Report
This is an interesting paper looking at change in social functions and loneliness during the COVID-19 outbreak in a large sample of Belgian gay men and men who have sex with men. Authors tested a series of hypotheses finding diverse and complex results. I think that the paper deserves to be published but that authors should make a big effort to make it more comprehensible and linear. Indeed, the paper does not follow the traditional sections of a scientific paper, and this makes hard to follow the complex discourse. Following, I report my recommendations:
INTRODUCTION
I suggest authors to reorganize the whole Introduction using subheadings. First, authors can include a very short initial paragraph stating the main problem addressed and the objectives. Then, they can present the literature on social isolation during the COVID-19 lived by both general population and LGBTQ+ people, highlighting the specific challenges experienced by this specific population. Furthermore, being the sample recruited in Belgium, it could be useful to readers to know something about the socio-cultural context in which LGBTQ+ people live and how the COVID-19 outbreak impacted on life of Belgian LGBTQ+ people. Another paragraph should include the theoretical framework that authors included in Materials and Methods. Indeed, theoretical consideration should not be included in methods section. Finally, a last paragraph should present the hypotheses. Summarizing, my suggestion is to reorganize the introduction as follows: (1) main problems and general objective; (2) Social isolation and COVID-19 in general population and LGBTQ+ people; (3) theoretical frameworks about loneliness; (4) the current study (specific objectives and hypotheses + Figure 1).
METHODS
This section has the same problems as the Introduction. I think that also this section should be reorganized. I suggest authors to follow the traditional subheadings, as follows: (1) Participants; (2) Procedures; (3) Measures; (4) Statistical analyses.
Regarding measures, authors can use further subheadings, one for each measure (including socio-demographic characteristics that are not reported). Furthermore, authors should report statistics about the reliability and validity of the scale. Additionally, I suggest reporting some statistics about the power of the sample, as they performed many analyses which would require many participants.
RESULTS
This section is confusing. First, titles of the subparagraphs should not report the type of analysis (e.g., “linear regression”) but the main concepts analyzed (e.g., “associations between ….”). Even tables are quite confusing. Maybe, adapting them to the editorial norms can make them clearer. Why the p-value is reported in a separated column with asterisks? Asterisks should be reported on beta coefficients. If authors prefer to report p-values in a separated column, they should report the exact p-values.
DISCUSSION
Same problem as before. Authors started the discussion by highlighting limitations of the study (which should be reported at the end). Being many different results associated with 10 hypotheses, I suggest authors to reorganize the whole discussion in subheadings. At the end of the discussion, authors can report limitations and then social, clinical, and public health implications of their study.
Round 2
Reviewer 1 Report
Strengths of the study include the investigation of an important, timely topic in a vulnerable population.
The authors have done a fine job with the revision and the manuscript has improved considerably as a result. A few minor additional edits are needed:
Alpha rather than "alfa" should be used throughout.
Line 98 should be "where" our study takes place rather than "were".
Line 110 should be "report" about 2020 rather than "rapport"
Reviewer 4 Report
I complimenti with authors for their excellent revisions. I do not have other points to address.
Author Response
Dear reviewer,
Thank you very much for your compliment on the revision, and your thoughtful feedback in round 1 which made the improvement possible.
Sincerely,
Estrelle